# Estimating the Unique Information of Continuous Variables

**Ari Pakman**
Columbia University

**Amin Nejatbakhsh**
Columbia University

**Dar Gilboa**
Harvard University

**Abdullah Makkeh**
Georg August University

**Luca Mazzucato**
University of Oregon

**Michael Wibral**
Georg August University

**Elad Schneidman**
Weizmann Institute

## Abstract

The integration and transfer of information from multiple sources to multiple targets is a core motive of neural systems. The emerging field of partial information decomposition (PID) provides a novel information-theoretic lens into these mechanisms by identifying synergistic, redundant, and unique contributions to the mutual information between one and several variables. While many works have studied aspects of PID for Gaussian and discrete distributions, the case of general continuous distributions is still uncharted territory. In this work we present a method for estimating the unique information in continuous distributions, for the case of one versus two variables. Our method solves the associated optimization problem over the space of distributions with fixed bivariate marginals by combining copula decompositions and techniques developed to optimize variational autoencoders. We obtain excellent agreement with known analytic results for Gaussians, and illustrate the power of our new approach in several brain-inspired neural models. Our method is capable of recovering the effective connectivity of a chaotic network of rate neurons, and uncovers a complex trade-off between redundancy, synergy and unique information in recurrent networks trained to solve a generalized XOR task.

## 1 Introduction and background

In neural systems, often multiple neurons are driven by one external event or stimulus; conversely multiple neural inputs can converge onto a single neuron. A natural question in both cases is how multiple variables hold information about the singleton variable. In their seminal work [1], Williams and Beer proposed an axiomatic extension of classic information theory to decompose the mutual information between multiple source variables and a single target variable in a meaningful way. For the case of two sources $X_1, X_2$, their partial information decomposition (PID) amounts to expressing the mutual information of $X_1, X_2$ with a target $Y$ as a sum of four non-negative terms,

$$I(Y : (X_1, X_2)) = U(Y : X_1 \backslash X_2) + U(Y : X_2 \backslash X_1) + R(Y : (X_1, X_2)) + S(Y : (X_1, X_2)), \quad (1.1)$$

corresponding to unique ($U_1$, $U_2$), redundant ($R$) and synergistic ($S$) contributions, respectively. These terms should also obey the consistency equations

$$I(Y : X_1) = R(Y : (X_1, X_2)) + U(Y : X_1 \backslash X_2), \quad (1.2)$$

$$I(Y : X_2) = R(Y : (X_1, X_2)) + U(Y : X_2 \backslash X_1). \quad (1.3)$$

The PID has proved useful in understanding information processing by distributed systems in a diverse array of fields including machine learning [2, 3], earth science [4] and cellular automata [5],

35th Conference on Neural Information Processing Systems (NeurIPS 2021).

and particularly in neuroscience [6–10], where notions of synergy and redundancy, traditionally considered mutually exclusive and distinguished by the sign of

$$
\begin{aligned}
\Delta &= I(Y:(X_1,X_2)) - I(Y:X_1) - I(Y:X_2)\,, \\
&= S(Y:(X_1,X_2)) - R(Y:(X_1,X_2))\,,
\end{aligned}
\tag{1.4}
$$

have long played a central role in the quest to understand how neural circuits integrate information from multiple sources [11–14]. The novelty of the PID framework here is in separating the measures of synergy and redundancy in (1.4).

The above abstract formulation of PID provides three equations for four unknowns, and only becomes operational once one of $U_1$, $U_2$, $R$, or $S$ is defined. This has been done in [15] via a definition of the unique information:

**Definition 1** (BROJA [15]). *Given three random variables $(Y, X_1, X_2)$ with joint probability density $p(y, x_1, x_2)$, the unique information $U_1$ of $X_1$ with respect to $Y$ is*

$$
U(Y:X_1\backslash X_2) = \min_{q\in Q} I_q(Y:X_1|X_2)\,,
\tag{1.5}
$$

$$
= \min_{q\in Q} \int dy\,dx_1\,dx_2\, q(y,x_1,x_2) \log\left(\frac{q(y,x_1|x_2)}{q(y|x_2)q(x_1|x_2)}\right)\,,
\tag{1.6}
$$

*where*

$$
Q = \{q(y,x_1,x_2)\,|\,q(y,x_i) = p(y,x_i), i = 1, 2\}\,.
\tag{1.7}
$$

In words, we minimize the conditional mutual information $I(Y:X_1|X_2)$ over the space of density functions that preserve the marginal densities $p(y,x_1)$ and $p(y,x_2)$. The above definition implies, along with (1.2)-(1.3), that the unique and redundant information only depend on the marginals $p(y,x_1), p(y,x_2)$, and that the synergy can only be estimated from the full $p(y,x_1,x_2)$.

The original definition in [15] was limited to discrete random variables. Here, we show that the extension to continuous variables is well-defined and can be practically estimated.

**Motivation from decision theory** [15]. Consider for simplicity discrete variables. A decision maker $DM_1$ can choose an action $a$ from a finite set $\mathcal{A}$, and receives a reward $u(a,y)$ based on the selected action and the state $y$, which occurs with probability $p(y)$. Notably, $DM_1$ has no knowledge of $y$, but observes instead a random signal $x_1$ sampled from $p(x_1|y)$. Choosing the action maximizing the expected reward for each $x_1$, his maximal expected reward is

$$
R_1 = \sum_{x_1} p(x_1) \max_{a|x_1} \sum_y p(y|x_1)u(a,y)\,.
\tag{1.8}
$$

$DM_1$ is said to have no unique information about $y$ w.r.t. another decision maker $DM_2$ that observes $x_2 \sim p(x_2|y)$ – if $R_2 \geq R_1$ for any set $\mathcal{A}$, any distribution $p(y)$, and any reward function $u(a,y)$. A celebrated theorem by Blackwell [16, 17] states that such a generic advantage by $DM_2$ occurs iff there exist a stochastic matrix $q(x_1|x_2)$ which satisfies

$$
p(x_1|y) = \sum_{x_2} p(x_2|y)q(x_1|x_2)\,.
\tag{1.9}
$$

But this occurs precisely when the unique information (1.5) vanishes, since then there exists a joint distribution $q(y,x_1,x_2)$ in $Q$ for which $y \perp x_1|x_2$, which implies $q(x_1|x_2,y) = q(x_1|x_2)$, and thus (1.9) holds. Similar results exist for continuous variables [18, 19]. Thus the unique information from Definition 1 quantifies a departure from Blackwell's relation (1.9).

In this work we present a definition and a method to estimate the BROJA unique information for generic continuous probability densities. Our approach is based on the observation that the constraints (1.7) can be satisfied with an appropriate copula parametrization, and makes use of techniques developed to optimize variational autoencoders. We only consider one-dimensional $Y, X_1, X_2$ for simplicity, but the method can be naturally extended to higher dimensional cases. In Section 2 we review related works, in Section 3 we present our method and Section 4 contains several illustrative examples.

## 2    Related works

Partial information decomposition offers a solution to a repeated question that was not addressed by 'classical' information theory regarding the relations between two sources and a target [1]. From a mathematical perspective a 'functional definition' has to be made, meaning that such a definition should align with our intuitive notions. Yet, as shown in [20], not all intuitively desirable properties of a PID can be realized simultaneously. Thus, different desirable properties are chosen for distinct application scenarios. Thus, various proposals for decomposition measures are not seen as conflicting but as having different operational interpretations. For example, the BROJA approach used here builds on desiderata from decision theory, while other approaches appeal to game theory [21] or the framework of Kelly gambling [22]. Yet other approaches use arguments from information geometry [23]. Other approaches assume agents receiving potentially conflicting or incomplete information about the source variables for the purpose of inference or decryption (see e.g. [24, 25]). In [26] the authors separate the specific operational interpretations of PID measures from the general structure of information decomposition.

The actual computation of the BROJA unique information is non-trivial, even for discrete variables. Optimization methods exist for the latter case [27–29], and analytic solutions are only known when all the variables are univariate binary [30]. For continuous probability densities, an earlier definition aligned with the BROJA measure was made by Barret [31], but only applies to Gaussian variables. For Barret's measure, an analytic solution is known when $p(y, x_1, x_2)$ is a three-dimensional Gaussian density [31], but does not generalize to higher dimensional Gaussians [32].

## 3    Bounding and estimating the unique information

We proceed in two steps. We first introduce a parametrization of the optimization space $Q$ in (1.7) and then introduce and optimize an upper bound on the unique information.

### 3.1    Parametrizing the optimization space with copulas

To characterize the optimization space $Q$ in (1.5)-(1.7), it is convenient to recall that according to Sklar's theorem [33], any $n$-variate probability density can be expressed as

$$p(x_1 \dots x_n) = p(x_1) \dots p(x_n) c(u_1 \dots u_n) \,, \tag{3.1}$$

where $p(x_i)$ is the marginal and $u_i = F(x_i)$ is the CDF of each variable. The dependency structure among the variables is encoded in the function $c{:}[0,1]^n \to [0,1]$. This is a *copula* density, a probability density on the unit hypercube with uniform marginals [34],

$$\int_{[0,1]^{n-1}} \prod_{j=1, j \neq i}^{n} du_j \, c(u_1 \dots u_n) = 1 \quad \forall i \,. \tag{3.2}$$

Note that under univariate reparametrizations $z_i' = g(z_i)$, the $u_i$'s and the copula $c$ remain invariant. For an overview of copulas in machine learning, see [35].

**Proposition 1.** *Under the BROJA Definition 1 of unique information, all the terms of the partial information decomposition in (1.1)-(1.3) are independent of the univariate marginals $p(x_1), p(x_2), p(y)$, and only depend on the copula $c(u_y, u_1, u_2)$.*

*Proof.* Expressing $q(y, x_1, x_2), q(x_1, x_2), q(y, x_2)$ via copula decompositions (3.1), and changing variables as $du_y = q(y)dy$, etc., the objective function in (1.6) becomes

$$I_q(Y : X_1 | X_2) \quad = \quad \int_{[0,1]^3} du_y du_1 du_2 \, c(u_y, u_1, u_2) \log \left( \frac{c(u_y, u_1, u_2)}{c(u_y, u_2)c(u_1, u_2)} \right) \,. \tag{3.3}$$

Note that the copula of any marginal distribution is the marginal of the copula:

$$c(u_y, u_2) = \int_{[0,1]} du_1 \, c(u_y, u_1, u_2) \,, \qquad\qquad c(u_1, u_2) = \int_{[0,1]} du_y \, c(u_y, u_1, u_2) \,. \qquad (3.4)$$

Thus the optimization objective and the unique information are independent of the univariate marginals. A similar result holds for the mutual information terms in the l.h.s. of (1.1)-(1.3).[1] It follows that none of the PID terms in (1.1)-(1.3) depend on the univariate marginals, and therefore all the PID terms are invariant under univariate reparametrizations of $(y, x_1, x_2)$. □

In order to parametrize the optimization space $Q$ in (1.7) using copulas, consider the factorization

$$p(y, x_1, x_2) = p(x_1)p(y|x_1)p(x_2|y, x_1) \,. \qquad (3.5)$$

Using the copula decomposition (3.1) for $n = 2$, the last two factors in (3.5) can be expressed as

$$p(y|x_1) \quad = \quad \frac{p(y, x_1)}{p(x_1)} = \frac{p(y)p(x_1)c(y, x_1)}{p(x_1)} = c(u_y, u_1)p(y) \,, \qquad (3.6)$$

and similarly

$$p(x_2|y, x_1) \quad = \quad \frac{p(x_1, x_2|y)}{p(x_1|y)} \,, \qquad (3.7)$$

$$= \quad c_{1,2|y}(u_{1|y}, u_{2|y})p(x_2|y) \,, \qquad (3.8)$$

$$= \quad c_{1,2|y}(u_{1|y}, u_{2|y})c(u_y, x_2)p(x_2) \,, \qquad (3.9)$$

where we defined the conditional CDFs,

$$u_{i|y} = F(u_i|u_y) = \frac{\partial C(u_y, u_i)}{\partial u_y} \qquad i = 1, 2 \qquad (3.10)$$

and $C(u_y, u_i)$ is the CDF of $c(u_y, u_i)$. Note that the function $c_{1,2|y}(u_{1|y}, u_{2|y})$ in (3.8) is not the conditional copula $c(u_1, u_2|u_y)$, but rather the copula of the conditional $p(x_1, x_2|y)$. Using expressions (3.6) and (3.9), the full density (3.5) becomes

$$p(y, x_1, x_2) \quad = \quad p(y)p(x_1)p(x_2)c(u_y, u_1, u_2) \,, \qquad (3.11)$$

where

$$c(u_y, u_1, u_2) = c(u_y, u_1) \, c(u_y, u_2)c_{1,2|y}(u_{1|y}, u_{2|y}) \,. \qquad (3.12)$$

This is a simple case of the pair-copula construction of multivariate distributions [38–40], which allows to expand any $n$-variate copula as a product of (conditional) bivariate copulas.

**Proposition 2.** *The copula of the conditional, $c_{1,2|y}(\cdot, \cdot)$, parametrizes the space $Q$ in (1.7).*

*Proof.* Since $q(y, x_i) = p(y, x_i)$ $(i = 1, 2)$, the copula factors in

$$p(y, x_i) = p(y)p(x_i) \, c(u_y, u_i) \,, \qquad i = 1, 2 \qquad (3.13)$$

are fixed in $Q$. Therefore, in the copula decomposition (3.12) for $q(y, x_1, x_2) \in Q$, only the last factor can vary in $Q$. Let us denote by $\theta$ the parameters of a generic parametrization for the copula $c_{1,2|y}(u_{1|y}, u_{2|y})$. Since the latter is conditioned on $u_y$, the parameters can be taken as a function $\theta(u_y)$. It follows that the copula of $q$ necessarily has the form

$$c_\theta(u_y, u_1, u_2) = c(u_y, u_1) \, c(u_y, u_2) \, c_{1,2|\theta(u_y)}(u_{1|y}, u_{2|y}) \,, \qquad (3.14)$$

and the parameters of the function $\theta(u_y)$ are the optimization variables.[2] □

---

[1] The connection between mutual information and copulas was discussed in [36, 37].

[2] We note that in multivariate pair-copula expansions it is common to assume constant conditioning parameters $\theta$ [41], but we do not make such a simplifying assumption.

## 3.2 Optimizing an upper bound

Inserting now the expression (3.14) into the objective function (3.3) we get

$$I[\theta] = \mathbb{E}_{c_\theta(u_y,u_1,u_2)} \log \left[ c(u_y,u_1) c_{1,2|\theta(u_y)}(u_{1|y},u_{2|y}) \right] - \mathbb{E}_{c_\theta(u_1,u_2)} \log c_\theta(u_1,u_2), \quad (3.15)$$

which is our objective function and satisfies the marginal constraints (1.7). Note that apart from the optimization parameters $\theta$, it depends on the bivariate copulas $c(u_y,u_1)$ and $c(u_y,u_2)$ which should be estimated from the observed data. Given $D$ observations $\{y^{(i)}, x_1^{(i)}, x_2^{(i)}\}_{i=1}^D$, we map each value to $[0,1]$ via the empirical CDFs of each coordinate $(y, x_1, x_2)$. Computing the latter has a $O(D \log D)$ cost from sorting each coordinate and yields a data set $\{u_y^{(i)}, u_1^{(i)}, u_2^{(i)}\}_{i=1}^D$. The latter set is used to estimate copula densities $c(u_y,u_1)$ and $c(u_y,u_2)$ by fitting several parametric and non-parametric copula models [42], and choosing the best pair of models using the AIC criterion.[3] From the learned copulas we also get the conditional CDF functions $u_{i|y} = F(u_i|u_y)$ that appear in the arguments of the first term in (3.15).

**A variational upper bound.** Minimizing (3.15) directly w.r.t. $\theta$ is challenging because the second term depends on the copula marginal $c_\theta(u_1,u_2)$ which has no closed form, as it requires integrating (3.14) w.r.t. $u_y$. We introduce instead an inference distribution $r_\phi(u_y|u_1,u_2)$, with parameters $\phi$, that approximates the conditional copula $c_\theta(u_y|u_1,u_2)$, and consider the bound

$$\log c_\theta(u_1,u_2) = \log \int du_y' \, c_\theta(u_y',u_1,u_2) \geq \int du_y' \, r_\phi(u_y'|u_1,u_2) \log \frac{c_\theta(u_y',u_1,u_2)}{r_\phi(u_y'|u_1,u_2)}, \quad (3.16)$$

which follows from Jensen's inequality and is tight when $r_\phi(u_y'|u_1,u_2) = c_\theta(u_y'|u_1,u_2)$. This expression gives an upper bound on $I_q[\theta]$, which can be minimized jointly w.r.t. $(\theta, \phi)$.

A disadvantage of the bound (3.16) is that its tightness depends strongly on the expressiveness of the inference distribution $r_\phi(u_y'|u_1,u_2)$. This situation can be improved by considering a multiple-sample generalization proposed by [44],

$$\log c_\theta(u_1,u_2) \geq D_{A,\theta,\phi}(u_1,u_2) \equiv \mathbb{E}_{p(u_y^{(1)}...u_y^{(A)})} \log \left[ \frac{1}{A} \sum_{a=1}^A \frac{c_\theta(u_y^{(a)},u_1,u_2)}{r_\phi(u_y^{(a)}|u_1,u_2)} \right], \quad (3.17)$$

where the expectation is w.r.t. $A$ independent samples of $r_\phi(u_y'|u_1,u_2)$. $D_{A,\theta,\phi}(u_1,u_2)$ coincides with the lower bound in (3.16) for $A = 1$ and satisfies [44]

$$D_{A+1,\theta,\phi}(u_1,u_2) \geq D_{A,\theta,\phi}(u_1,u_2), \quad (3.18)$$

$$\lim_{A \to \infty} D_{A,\theta,\phi}(u_1,u_2) = \log c_\theta(u_1,u_2). \quad (3.19)$$

Thus, even when $r_\phi(u_y'|u_1,u_2) \neq c_\theta(u_y'|u_1,u_2)$, the bound can be made arbitrarily tight for large enough $A$. Inserting (3.17) in (3.15), we get finally

$$I_q[\theta] \leq B_1[\theta] + B_2[\theta, \phi], \quad (3.20)$$

where

$$B_1[\theta] = \mathbb{E}_{c_\theta(u_y,u_1,u_2)} \log \left[ c(u_y,u_1) c_{1,2|\theta(u_y)}(u_{1|y},u_{2|y}) \right], \quad (3.21)$$

$$B_2[\theta,\phi] = -\mathbb{E}_{c_\theta(u_1,u_2)} D_{A,\theta,\phi}(u_1,u_2), \quad (3.22)$$

and we minimize the r.h.s. of (3.20) w.r.t. $(\theta, \phi)$. Low-variance estimates of the gradients to perform the minimization can be obtained with the reparametrization trick [45, 46], as discussed in detail in Appendix A. In our examples below we use for $c_{1,2|\theta(u_y)}$ a bivariate Gaussian copula (reviewed in Appendix B). Such a copula has just one parameter $\theta \in [-1, +1]$, and thus the optimization is done over the space of functions $\theta(u_y):[0,1] \to [-1,+1]$, which we parametrize with a two-layer neural network. Similarly, we parametrize $r_\phi(u_y|u_1,u_2)$ with a two-layer neural network. Details of these networks are in Appendix D.

---

[3]For this fitting/model selection step, we used the `pyvinecopulib` python package [43].

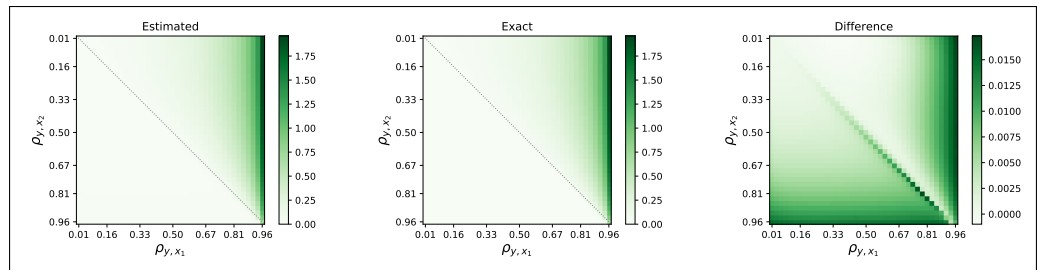

Figure 1: **Estimated vs. exact values of unique information for Gaussians.** For a three-dimensional Gaussian, we show estimates of $U(Y : X_1 \backslash X_2)$ as a function of the correlations $\rho_{y,x_i} (i = 1, 2)$, compared with the exact results from [31]. Only for Gaussian distributions are exact results known for continuous variables.

While the term $B_2$ in our bound is similar to the negative of the ELBO bound in importance weighted autoencoders (IWAEs) [44], there are some differences between the two settings, the most important being that we are interested in the precise value of the bound at the minimum, rather than the learned functions $c_\theta, r_\phi$. Note also that our latent variables $u_y^{(k)}$ are one-dimensional, as opposed to the usual higher dimensional latent distributions of variational autoencoders, and that the empirical expectation over data observations in IWAEs is replaced in $B_2$ by the expectation over $c_\theta(u_1, u_2)$, whose parameters are also optimized.

**Estimating the other PID terms** In the following we adopt the minimal value taken by the upper bound (3.20) as our estimate of $U_1$. The other terms in the partial information decomposition are obtained from the consistency relations (1.1)-(1.3), after estimating the mutual informations $I(Y : (X_1, X_2)), I(Y : X_1), I(Y : X_2)$. There are several methods for the latter. In our examples, we use the observed data to fit additional copulas $c(u_1, u_2)$ and $c_{12|\theta(u_y)}$ and estimate $I(Y : X_1) \simeq \frac{1}{D} \sum_{i=1}^{D} \log c(u_y^{(i)}, u_1^{(i)})$ and similarly for the other terms. Note that all our estimates have sources of potential bias. Firstly, the estimation of the parametric copulas is subject to model or parameter misspecification, which can be ameliorated by more refined model selection strategies. Secondly, the optimized bound might not saturate, biasing the estimate upwards. This can be improved using higher $A$ values and improving the gradient-based optimizer used.

## 4    Examples

**Comparison with exact results for Gaussians.** Consider a three-dimensional Gaussian with correlations $\rho_{y,x_i}$ between $y, x_i$ for $i = 1, 2$. The exact solution to (1.5) in this case is [31]

$$U(Y : X_1 \backslash X_2) = \frac{1}{2} \log \left( \frac{1 - \rho_{y,x_2}^2}{1 - \rho_{y,x_1}^2} \right) \mathbb{1} \left[ \rho_{y,x_2} < \rho_{y,x_1} \right]. \tag{4.1}$$

Fig. 1 compares the above expression with estimates from our method. Here we know that $c_{y,1}$ and $c_{y,2}$ are Gaussian copulas, with parameters $\rho_{y,x_1}, \rho_{y,x_2}$, and we assumed a Gaussian copula for $c_{1,2|y,\theta}(u_{1|y}, u_{2|y})$ as well. For each pair of values $\rho_{y,x_1}, \rho_{y,x_2}$. In this and the rest of the experiments, we optimized the parameters $(\theta, \phi)$ using the ADAM algorithm [47] with a fixed learning rate $10^{-2}$ during 1200 iterations, and using $A = 50$. The results reported correspond to the mean of the bound in the last 100 iterations. The comparison in Fig. 1 shows excellent agreement.

**Model systems of three neurons.** The nature of information processing of neural systems is a prominent area of application of the PID framework, since synergy has been proposed as natural measure of information modification [7, 48]. We consider two models:

$$\begin{array}{cc} \mathbf{M1} & \mathbf{M2} \\ (X_1, X_2) \sim \mathcal{N}(0, \rho_{12}^2), & (X_1, X_2) \sim \mathcal{N}(0, \rho_{12}^2), \\ Y = \tanh(w_1 X_1 + w_2 X_2). & Y = X_1^2 / \left( 0.1 + w_1 X_1^2 + w_2 X_2^2 \right). \end{array} \tag{4.2}$$

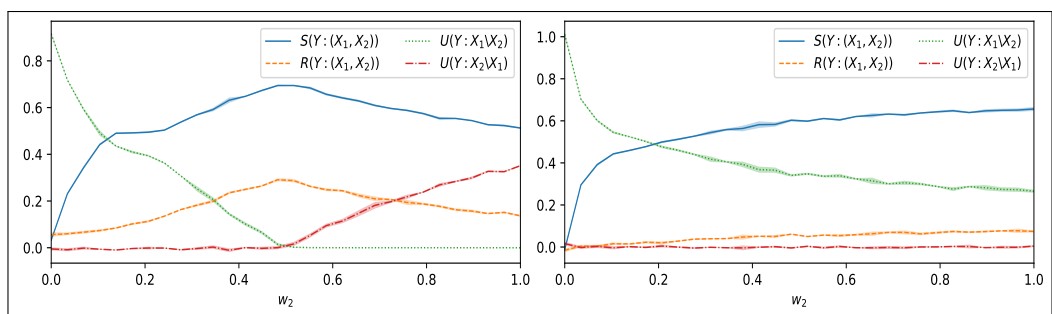

Figure 2: **Partial information decomposition for two neural network models.** In both models (4.2) we fixed $w_1 = 0.5, \rho_{12} = 0.3$, and show the PID terms as a function of the synaptic strength $w_2$, normalized by $I(Y : (X_1, X_2))$. We show mean (lines) and standard deviations (shaded area around each line) from 3 runs. *Left:* Model 1: The input of greatest weight conveys all the unique information, and synergy and redundancy both peak as $w_1 = w_2$. *Right:* Model 2: The second input $X_2$ has negligible unique information contribution, but its synaptic strength $w_2$ modulates the synergistic term, associated to the modification of information the neuron performs [48].

Both models are parameterized by the correlation $\rho_{12}$ and weights $w_1, w_2$. Model 1 is a particularly simple neural network. The tanh activation does not affect its copula, and even for a linear activation function the variables are not jointly Gaussian since $Y$ is deterministic on $(X_1, X_2)$. Model 2 is inspired by a normalization operation widely believed to be canonical in neural systems [49] and plays a role in common learned image compression methods [50]. The results, presented in Figure 2. are obtained from 3000 samples from each model

**Computational aspects of connectivity in recurrent neural circuits.** We apply our continuous variable PID to understand computational aspects of the information processing between recurrently coupled neurons (Fig. 3). A large amount of work has been devoted to applying information theoretic measures for quantifying directed pairwise information transfer between nodes in dynamic networks and neural circuits [51]. However, classical information theory only allows for the quantification of information transfer, whereas the framework of PID enables further decomposition of information processing into transfer, storage, and modification, providing further insights into the computation within a recurrent system [52]. Transfer entropy (TE) [53] is a popular measure to estimate the directed transfer of information between pairs of neurons [54, 55], and is sometimes approximated by linear Granger causality. Intuitively, TE between a process $X$ and a process $Y$ measures how much the past of $X$, $X^-$, can help to predict the future of $Y$, $Y^+$, accounting for its past $Y^-$. Although TE quantifies how much information is transferred between neurons, it does not shed light on the computation emerging from the interaction of $X^-$ and $Y^-$. Simply put, the information transferred from $X^-$ could enter $Y^+$, independently of the past state $Y^-$, or it could be fused in a non-trivial way with the information in the state in $Y^-$ [52, 56]. PID decomposes the TE into **modified transfer** (quantified by $S(Y^+:X^-, Y^-)$) and **unique transfer** (quantified by $U(Y^+:X^- \setminus Y^-)$) terms (see the Appendix for a proof):

$$TE(X \rightarrow Y) = I(Y^+:X^-|Y^-) = U(Y^+:X^- \setminus Y^-) + S(Y^+:X^-, Y^-).$$

Furthermore, the information kept by the system through time can be quantified by the **unique storage** (given by $U(Y^+:Y^- \setminus X^-)$) and **redundant storage** (given by $R(Y^+:X^-, Y^-)$) in PID [48]. This perspective is a new step towards understanding how the information is processed in recurrent systems beyond merely detecting the direction functional interactions estimated by traditional TE methods (see Appendix G, for details). To explore these ideas, we simulated chaotic networks of rate neurons with an a-priori causal structure consisting of two sub-networks $\mathbf{X}$ and $\mathbf{Y}$ (Fig. 3a, see [57] for more details on causal analyses of this network model). The sub-network $\mathbf{X}$ is a Rossler attractor of three neurons obeying the dynamical equations:

$$\begin{cases} \dot{X}_1 = -X_2 - X_3 \\ \dot{X}_2 = X_1 + \alpha X_2 \\ \dot{X}_3 = \beta + X_3(X_1 - \gamma) \end{cases} \tag{4.3}$$

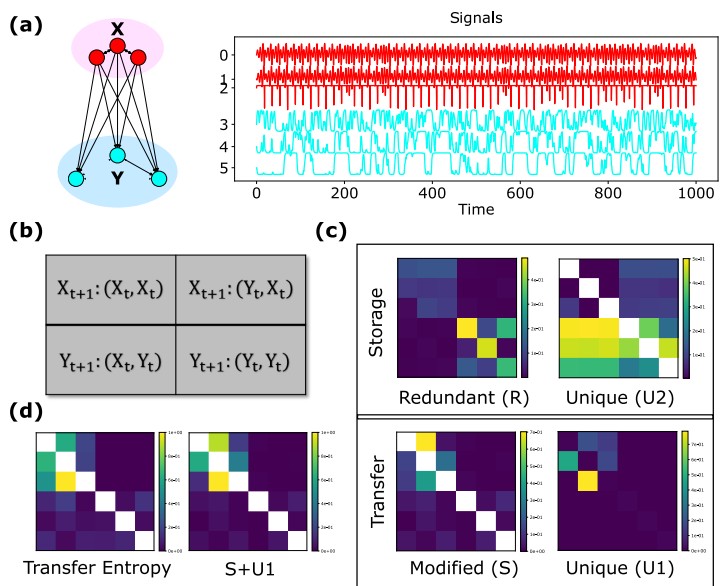

Figure 3: **PID uncovers the effective connectivity and allows for the quantification of storage, modification, and transfer of information in a chaotic network of rate neurons.** **a**: Schematics of recurrent network architecture (left) and representative activity (right). **b**: Schematic of the PID triplets for each $3 \times 3$ block of the matrices in c, d. **c:** PID decomposition into modified transfer $S$, unique transfer $U_1$, redundant storage $R$, and unique storage $U_2$ for the rate network. The future of $X$ neurons only depends on unique information in the past of $X$ neurons and their synergistic interactions. The interactions between the $X$ and $Y$ sub-networks only contain synergistic information regarding the future of $Y$ but no redundant information; the latter is only present in the interactions confined within each sub-network. **d**: The transfer entropy (TE), estimated via IDTxl [58], recovers the sum of modified and unique transfer terms $S + U_1$.

where $\{\alpha, \beta, \gamma\} = \{0.2, 0.2, 5.7\}$. There are 100 neurons in the sub-network **Y** from which we chose the first three, $Y_{1:3}$, to simulate the effect of unobserved nodes. Neurons within the sub-network $Y$ obey the dynamical equations

$$\dot{Y} = -\lambda Y + 10 \tanh(J_{YX} X + J_{YY} Y) \tag{4.4}$$

where $J_{YX} \in \mathbb{R}^{100 \times 3}$ has all its entries equal to $0.1$, and $J_{YY}$ is the recurrent weight matrix of the $Y$ sub-network, sampled as zero-mean, independent Gaussian variables with standard deviation $g = 4$. No projections exist from the downstream sub-network **Y** to the upstream sub-network **X**. We simulated time series from this network (exhibiting chaotic dynamics, see Fig. 3a) and estimated the PID as unique, redundant, and synergistic contribution of neuron $i$ and neuron $j$ at time $t$ in shaping the future of neuron $j$ at time $t + 1$. For each pair of neurons $Z_i, Z_j \in \{X_{1:3}, Y_{1:3}\}$ we treated $(Z_i^t, Z_j^t, Z_j^{t+1})_{t=1}^{T}$ as iid samples[4] and ran PID on these triplets ($i, j$ represent rows and columns in Fig. 3b-d). The PID uncovered the functional architecture of the network and further revealed non-trivial interactions between neurons belonging to the different sub-networks, encoded in four matrices: modified transfer $S$, unique transfer $U_1$, redundant storage $R$, and unique storage $U_2$ (details in Fig. 3d). The sum of the modified and unique transfer terms was found to be consistent with the TE (Fig. 3c, TE equal to $S + U_1$, up to estimation bias). The TE itself captured the network effective connectivity, consistent with previous results [55, 57].

**Uncovering a plurality of computational strategies in RNNs trained to solve complex tasks.** A fundamental goal in neuroscience is to understand the computational mechanisms emerging from

---

[4]Note that the estimation of the PID from many samples of the triplets $(Z_i^t, Z_j^t, Z_j^{t+1})$ is operationally the same whether such triplets are iid or, as in our case, temporally correlated. This is similar to estimating expectations w.r.t. the equilibrium distribution of a Markov chain by using temporally correlated successive values of the chain. In both cases, the temporal correlations do not introduce bias in the estimator but can increase the variance.

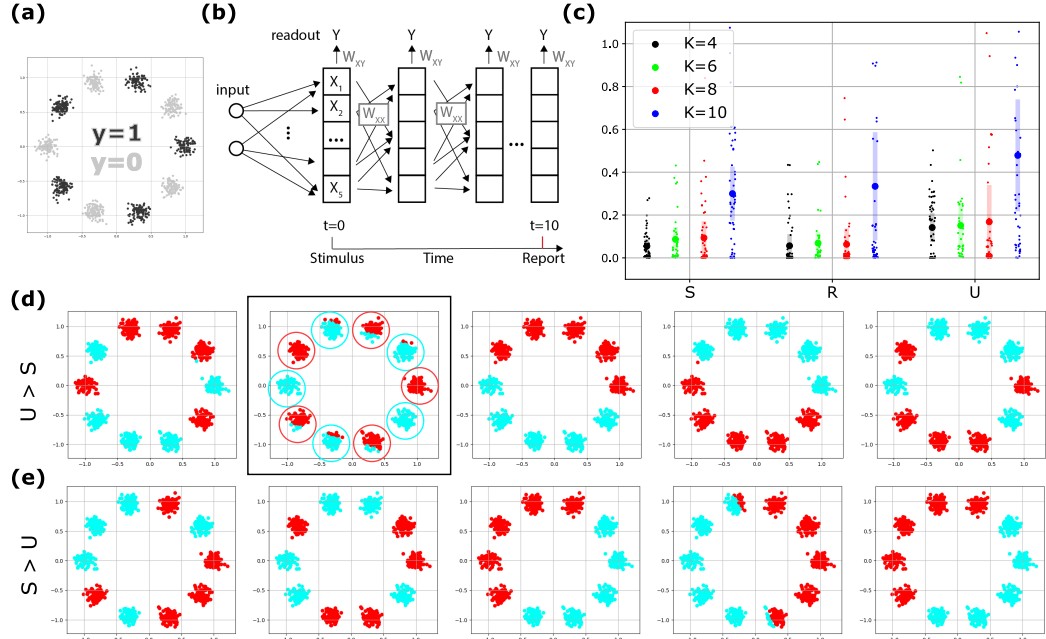

Figure 4: **PID of RNNs trained to solve generalized XOR problem. a**: Input data drawn from a 2D Gaussian Mixture Model with $K$ mixture components $X \sim \sum_{k=1}^{K} \frac{1}{K} \mathcal{N}(X|\mu_k, \sigma I)$ with means lying on the unit circle (grey and black dots represent the two class labels). **b**: Two layer network with 2D input layer, 5 recurrently connected hidden neurons $X$ and one readout neuron $Y$; RNN activity unfolds in time (horizontal axis). The input is presented at time $t = 0$, then withdrawn, and the RNN is trained with BPTT to report the decision at $t = 10$. In this representation, layers correspond to time-steps and weights $W_{XX}$ are shared between layers. **c**: PID between output $Y(t)$ and pairs of hidden neurons $X_i(t-1), X_j(t-1)$ for $t = 10$ yielding $S, R, U_1, U_2$ (distribution over 1000 input samples for each task $K$; 20 networks per task). Harder tasks led to an increase in PID measures. **d**: Example receptive fields for a network with $U > S$ shows emergence of grand-mother cells in the hidden layer (red and blue colors represent hidden neurons outputs; grandmother cell, second from left). **e**: Example receptive fields for a network with $S > U$, relying on higher synergy between neurons to solve the task.

the collective interactions of recurrent neural circuits leading to cognitive function and behavior. Here, we show that PID opens a new window for assessing how specific computations arise from recurrent neural interactions. Unlike MI or TE, the PID quantifies the alternative ways in which a neuron determines the information in its output from its inputs, and thus can be a sensitive marker of different computational strategies. We here trained RNNs as models of cortical circuits [59] and used the PID to elucidate how the computations emerging from recurrent neural interactions contribute to task performance. We trained RNNs to solve a generalized version of the classic XOR classification problem with target labels corresponding to odd vs. even mixture components (Fig. 4a). Stimuli were presented for one time step ($t = 0$) and the network was trained to report the decision at $t = 10$. By tracking the temporal trajectories of the hidden layer activity we found that the network recurrent dynamics (represented as unfolded in time in Fig. 4b) progressively pulls the two input classes in opposite directions along the output weights (see Appendix). We used PID to dissect how a plurality of different strategies emerge from recurrent neural interactions in RNNs trained for solving a classification task. The computation emerged from the recurrent interaction between hidden neurons at different time steps. Do all successfully trained networks have a similar profile in terms of the PID terms? If so, this hints at a single computational strategy across these networks. If not, it is safe to assume that task performance is reached via different mechanisms, despite identical network architecture and training algorithm.

We found that on average across multiple networks S, R, and U rose with task difficulty (Fig. 4c), yet at all difficulties, individual networks differed strongly with respect to the ratio $S/U$, i.e. there were networks with larger average synergy across neuron pairs compared to the average unique information, and vice versa. For simple networks like the ones used here, one can inspect receptive fields to understand the reason for this differential behaviour (Fig. 4d-e). Indeed, networks with high average unique information displayed 'grandmother-cell'-like neurons, that would alone classify a large parts of the sample space, while in networks with higher average synergy such cells were absent (Fig. 4d). The emergence of these 'grandmother-cell'-like receptive fields is due to the recurrent dynamics. While in a feedforward architecture ($W_{XX} = 0$) hidden layer receptive fields are captured by hyperplanes in input space, in the RNN the receptive fields are time dependent, where later times are interpreted as deeper layers (Fig. 4b) and thus can capture highly non-linear features in input space. The advantage of PID versus a manual inspection of receptive fields is twofold: First, the PID framework abstracts and generalizes descriptions of receptive fields as being e.g. 'grandmother-cell'-like; thus the concept of unique information stays relevant even in scenarios where the concept of a receptive field becomes meaningless, or inaccessible. Second, the quantitative outcomes of a PID rest only on information theory, not specfic assumptions about neural coding or computational strategies, and can be obtained for large numbers of neurons.

Comparison of our PID-based approach with the concept of neuronal selectivity used in neuroscience highlights interesting similarities and differences. Several kinds of selectivity (pure, mixed linear, and mixed non-linear) can be identified by performing regression analysis of neural responses vs. task variables [60]. In this framework, our grand-mother cells correspond to neurons with pure selectivity to the input class labels (a.k.a. "choice-selective" neurons). In the XOR task, [60] showed that non-linear mixed selectivity of neurons to the class labels is beneficial when solving the XOR task, by leading to a high-dimensional representation of the task variables. While selectivity profiles are a property of single neuron responses to task variables, our PID measures are a property of the combined activity of triplets of neurons and thus reveal emerging functional interactions between units and their computational algorithms (see also [7] and [52]). This allowed us to characterize a functional property of neural systems less studied than task variable selectivity: the computations that require functional mixing of the information from multiple units (measured by the average synergistic information) vs. the computations that rely on the output of individual neurons (measured by the unique information and described as grandmother cells). Concretely, by comparing PID and receptive fields we found that that in networks with high unique information, neurons typically have receptive fields with pure selectivity (grandmother cells, with large unique information to the class labels). In networks with high synergy, neurons show complex mixed selectivity to class labels.

# 5   Conclusions

We presented a partial information decomposition measure for continuous variables with arbitrary probability densities, thereby extending the popular BROJA PID measure for discrete variables. Extending PID measures to continuous variables drastically broadens the possible applications of the PID framework. This is important as the latter provides key insights into the way a complex system represents and modifies information in a computation – via asking which variables carry information about a target uniquely (such that it can only be obtained from that variable), redundantly, or only synergistically with other variables. Answering these questions is pivotal to understanding distributed computation in complex systems in general, and neural coding in particular. We believe that the methods presented here will allow PIDs to be extended efficiently in neuroscience for multiple continuous sources with potentially complex dependency structures, as would be common in cellular imaging data or activation properties of brain modules or areas in functional imaging. More generally, the approach we presented here would be relevant for other application domains such as machine learning, biomedical science, finance, and the physical sciences.

## Acknowledgments

We thank Thibault Vatter and Praveen Venkatesh for conversations. The work of AP is supported by the Simons Foundation, the DARPA NESD program, NSF NeuroNex Award DBI1707398 and The Gatsby Charitable Foundation. DG is supported by a Swartz Fellowship. AM and MW are supported by Volkswagenstiftung under the program 'Big Data in den Lebenswissenschaften' and by the Ministry for Science and Education of Lower Saxony and the Volkswagen Foundation through the 'Niedersächsisches Vorab'. LM is supported by NINDS Grant NS118461 (BRAIN Initiative). ES is supported by the Simons Collaboration on the Global Brain (542997) as well as research support from Martin Kushner Schnur and Mr. and Mrs. Lawrence Feis, and is the Joseph and Bessie Feinberg Professorial Chair.

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
