## Supplementary Material

## A  Estimating the gradients

As shown in Section 3, the unique information $U(Y : X_1 \backslash X_2)$ is upper bounded as

$$I_q[\theta] \leq B_1[\theta] + B_2[\theta, \phi] \,, \tag{A.1}$$

where

$$
\begin{aligned}
B_1[\theta] &= \mathbb{E}_{c_\theta(u_y, u_1, u_2)} \log \left[ c(u_y, u_1) c_{1,2|\theta(u_y)}(u_{1|y}, u_{2|y}) \right] \,, &\tag{A.2}\\
B_2[\theta, \phi] &= -\mathbb{E}_{c_\theta(u_1, u_2)} D_{A,\theta,\phi}(u_1, u_2) \,, &\tag{A.3}
\end{aligned}
$$

and

$$D_{A,\theta,\phi}(u_1, u_2) = \mathbb{E}_{r_\phi(u_y^{(1)} \ldots u_y^{(A)}|u_1,u_2)} \log \left[ \frac{1}{A} \sum_{a=1}^{A} \frac{c_\theta(u_y^{(a)}, u_1, u_2)}{r_\phi(u_y^{(a)}|u_1, u_2)} \right] \,, \tag{A.4}$$

and the above expectation is w.r.t.

$$r_\phi(u_y^{(1)} \ldots u_y^{(A)}|u_1, u_2) \equiv \prod_{a=1}^{A} r_\phi(u_y^{(a)}|u_1, u_2) \,. \tag{A.5}$$

The parametrization we use for the inference distribution $r_\phi(u_y|u_1, u_2)$ is detailed below in Appendix D. We are interested in minimizing the r.h.s. of (A.1) w.r.t. $(\theta, \phi)$. To obtain low-variance gradients, it is convenient to eliminate the $\theta, \phi$ dependence in the measures of (A.2)-(A.4) using the 'reparametrization trick' [45].

The idea is to obtain samples from $c_\theta(u_y, u_1, u_2)$ by a $\theta$-dependent transformation of three Unif$[0, 1]$ samples $\mathbf{v} = (v_y, v_1, v_2)$, and samples from $r_\phi(u_y|u_1, u_2)$ by a $(u_1, u_2, \phi)$-dependent transformation of $\epsilon \sim$ Unif$[0, 1]$. We present the details of these transformations in Appendices C and D, respectively.

Taking $M$ samples of $c_\theta(u_y, u_1, u_2)$ and denoting them as $\bar{u}_y^{(m)}, \bar{u}_1^{(m)}, \bar{u}_2^{(m)}$, we can estimate (A.2) as

$$B_1[\theta] \simeq \frac{1}{M} \sum_{m=1}^{M} \log \left[ c(\bar{u}_y^{(m)}, \bar{u}_1^{(m)}) \, c_{1,2|\theta\left(\bar{u}_y^{(m)}\right)}(\bar{u}_{1|y}^{(m)}, \bar{u}_{2|y}^{(m)}) \right] \tag{A.6}$$

where we denoted $\bar{u}_{i|y} = F(u_i = \bar{u}_i|u_y = \bar{u}_y)$ for $i = 1, 2$. An estimate of the gradient $\nabla_\theta B_1$ is obtained by acting on this expression with $\nabla_\theta$, which also acts on the $\theta$-dependent samples.

Denoting $A$ samples from $r_\phi(u_y|u_1, u_2)$ as $\hat{u}_y^{(a)}$, we can also estimate (A.3) as

$$B_2[\theta, \phi] \simeq -\frac{1}{M} \sum_{m=1}^{M} \log \left( \frac{1}{K} \sum_{a=1}^{A} w_{a,m} \right) \,, \tag{A.7}$$

where we defined

$$w_{a,m} = \frac{c_\theta(\hat{u}_y^{(a)}, \bar{u}_1^{(m)}, \bar{u}_2^{(m)})}{r_\phi(\hat{u}_y^{(a)}|\bar{u}_1^{(m)}, \bar{u}_2^{(m)})} \,. \tag{A.8}$$

Acting on this expressions with $\nabla_\theta$ yields an estimate of $\nabla_\theta B_2$. On the other hand, as noted in [61], the estimate of $\nabla_\phi B_2$ resulting from acting with $\nabla_\phi$ on (A.7) has a signal-to-noise ratio which decreases with $A$. A solution to this problem was found in [46], which showed that a stable gradient estimate can be obtained instead as

$$\nabla_\phi B_2 \simeq \frac{-1}{M} \sum_{m=1}^{M} \sum_{a=1}^{A} \left( \frac{w_{a,m}}{\sum_{s=1}^{A} w_{s,m}} \right)^2 \frac{\partial \log w_{a,m}}{\partial \hat{u}_y^{(a)}} \nabla_\phi \hat{u}_y^{(a)} \,, \tag{A.9}$$

and this is the estimate we use in our experiments.

## B    The bivariate Gaussian copula

A bivariate Gaussian copula is parametrized by $\theta \in [-1, 1]$ and given by

$$c(u_1, u_2) = \frac{1}{\sqrt{1 - \theta^2}} \exp\left\{ -\frac{\theta^2(x_1^2 + x_2^2) - 2\theta x_1 x_2}{2(1 - \theta^2)} \right\} \tag{B.1}$$

where $x_i = \Phi^{-1}(u_i)$ and $\Phi$ is the standard univariate Gaussian CDF. For explicit expressions of other popular bivariate copulas, see [39].

## C    Sampling from the copula

In this section we show how to obtain samples from the three-dimensional copula

$$c_\theta(u_y, u_1, u_2) = c(u_y, u_1)\, c(u_y, u_2)\, c_{1,2|y,\theta}(u_{1|y}, u_{2|y}) \tag{C.1}$$

by applying a $\theta$-dependent transformation to samples from $\mathrm{Unif}[0, 1]$. We use the Rosenblatt transform [62], which consists in using the inverse CDF method to sample from each factor in

$$c(u_y, u_1, u_2) = c(u_1)c(u_y|u_1)c(u_2|u_y, u_1). \tag{C.2}$$

We denote $F(\cdot|\cdot)$ is the CDF of $c(\cdot|\cdot)$. Adopting the notation of [39], we define

$$h_{ij}(u_i, u_j) = F(u_i|u_j) = \frac{\partial C(u_i, u_j)}{\partial u_j} \quad i, j = 1, 2. \tag{C.3}$$

For several popular parametric families of bivariate copulas, such as those we consider in this paper, explicit expressions are known for $h_{ij}(u_i, u_j)$ along with its inverse $h_{ij}^{-1}(\cdot, u_j)$ w.r.t. the first argument (see e.g. [39]). Note that using this notation, the arguments in the last factor of (C.1) are $u_{i|y} = h_{iy}(u_i, u_y)$ (i=1,2).

We first sample $(v_1, v_y, v_2)$ from $\mathrm{Unif}[0, 1]$ and successively obtain $u_1, u_y, u_2$ by inverting the functions in the r.h.s. of

$$
\begin{aligned}
v_1 &= F(u_1), \\
&= u_1, \\
v_y &= F(u_y|u_1), \\
&= h_{y1}(u_y, u_1), \\
v_2 &= F(u_2|u_1, u_y), \\
&= h_{21|\theta(u_y)}(F(u_2|u_y), F(u_1|u_y)), \\
&= h_{21|\theta(u_y)}(h_{2y}(u_2, u_y), h_{1y}(u_1, u_y)).
\end{aligned}
$$

Explicitly, we get

$$
\begin{aligned}
u_1 &= v_1, \\
u_y &= h_{y1}^{-1}(v_y, u_1), \\
u_2 &= h_{2y}^{-1}(h_{21|\theta(u_y)}^{-1}(v_2, h_{y1}(u_y, u_1)), u_y).
\end{aligned}
$$

Note that only $u_2$ actually depends on $\theta$.

# D  Parametrization of the learned models

## D.1  Parametrizing the learned conditional copula

The conditional Gaussian copula $c_{1,2|\theta(u_y)}(u_{1|y}, u_{2|y})$ is parametrized by the function $\theta(u_y){:}[0,1] \to [-1,+1]$. For its functional form we used

$$\theta(u_y) = \tanh\left(\sum_{i=1}^{16} w_{2,i} \tanh(w_{1,i} u_y + b_1) + b_2\right) \tag{D.1}$$

where $w_{1,i}, w_{2,i}, b_1, b_2 \in \mathbb{R}$.

## D.2  Parametrizing and sampling from the inference distribution

In our experiments we parametrize the inference distribution $r_\phi(u_y|u_1, u_2)$ via its CDF, as

$$\begin{aligned} R_\phi(u_y|u_1, u_2) &= \int_0^{u_y} du\, r_\phi(u|u_1, u_2)\,, \tag{D.2} \\ &= \frac{1}{1 + e^{-z(u_y)a_\phi(u_1,u_2) - b_\phi(u_1,u_2)}}\,, \end{aligned}$$

where $z(u_y) = \log\left(\frac{u_y}{1-u_y}\right)$. Derivating w.r.t. $u_y$ gives

$$r_\phi(u_y|u_1, u_2) = R_\phi(1 - R_\phi)a_\phi\left(u_y^{-1} + (1 - u_y)^{-1}\right)\,. \tag{D.3}$$

The functions $a_\phi(u_1, u_2)$ and $b_\phi(u_1, u_2)$ take values in $\mathbb{R}$ and are parametrized with a neural network with two hidden layers, and we impose $a_\phi(u_1, u_2) > 0$ in order to make $R_\phi$ monotonous with $u_y$. In order to sample from $r_\phi$, we draw $\epsilon \sim \text{Unif}[0,1]$, and use the inverse CDF method to obtain

$$\begin{aligned} u_y(\epsilon, u_1, u_2) &= R_\phi^{-1}(\epsilon|u_1, u_2)\,, \tag{D.4} \\ &= \frac{1}{1 + e^{-(z(\epsilon) - b_\phi(u_1,u_2))/a_\phi(u_1,u_2)}}\,. \end{aligned}$$

# E  Comparison with a discrete estimator

In this section we estimate the PID of the two models of three neurons from eq. (4.2) using the discrete estimator `BROJA-2PID` [29]. In particular, we present a quantization scheme of the continuous models that leads to a qualitative agreement between the discrete and continuous estimators, thus further validating the results of the latter.

Let us denote the discretized versions of $X_1, X_2, Y$ as $\hat{x}_1, \hat{x}_2, \hat{y}$. The discrete PID estimators require as input a distribution $p(\hat{x}_1, \hat{x}_2, \hat{y})$ [29]. To create the latter from our continuous models in eq. (4.2), we start by dividing the continuous range of each $X_i(i = 1, 2)$ into $N_x$ segments, and associate each segment with a discrete value $\hat{x}_i$ equal to the value of $X_i$ in the middle of each segment. To each square in the resulting 2D $N_x \times N_x$ grid we associate a discrete probability $p(\hat{x}_1, \hat{x}_2)$ equal to the integral of the joint Gaussian density of $(X_1, X_2)$ in the square. Finally, in each of the two models, we split the $Y$ range into $N_y$ segments $\{s_i\}_{i=1}^{N_y}$. The boundaries of the segments are chosen such that the same fraction $1/N_y$ of values of $Y = Y(X_1, X_2)$ falls into each segment using eq. (4.2), a procedure called 'maximum entropy binning'. Let $\hat{y} \in \{1 \dots N_y\}$. Using this quantization, the three-dimensional discrete distribution is defined as

$$p(\hat{x}_1, \hat{x}_2, \hat{y}) = \begin{cases} p(\hat{x}_1, \hat{x}_2) & \text{if } Y(\hat{x}_1, \hat{x}_2) \in s_{\hat{y}}\,, \\ 0 & \text{if } Y(\hat{x}_1, \hat{x}_2) \notin s_{\hat{y}}\,, \end{cases} \tag{E.1}$$

where in each model $Y(\hat{x}_1, \hat{x}_2)$ is obtained from eq. (4.2). Fig. 5 shows the results of the discrete PID obtained using this quantization for the two models considered, assuming $X_i \in [-8, 8]$ and $N_x = 16$ equally-sized segments. For the $Y$ quantization we used $N_y = 3$. Note the qualitative agreement with the continuous results in Fig. 2.

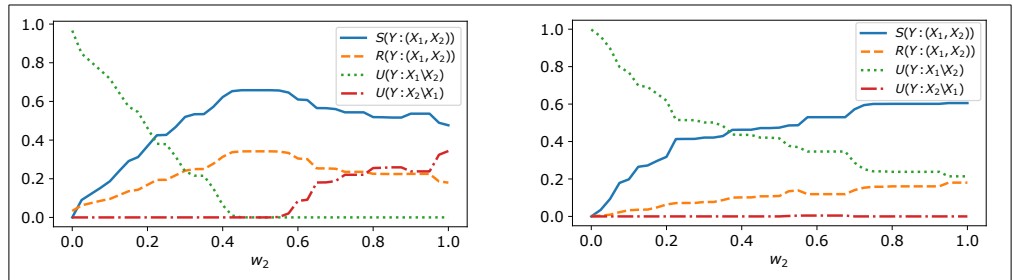

Figure 5: **Qualitative agreement of discrete and continuous PID estimations.** The two models are defined in (4.2) (left: Model 1, right: Model 2), and we used the same model parameters indicated in Fig. 2. We show the normalized discrete PID terms as a function of the synaptic strength $w_2$. See the text for details on the discrete quantization used. Note that for both models the discrete results agree qualitatively with the continuous results in Fig. 2.

# F  Consistency

Using Model 2 from Eq.(4.2) as an example, we compared estimates of $U(Y : X_2 \backslash X_1)$ with indirect estimates obtained from applying the consistency conditions to estimates of $U(Y : X_1 \backslash X_2)$. The results in Figure 6 show good agreement, thus further validating the method.

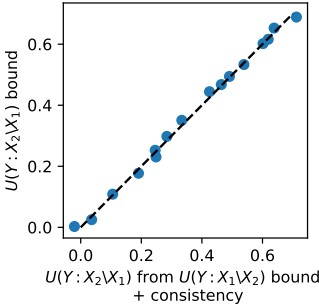

Figure 6: Comparison of direct vs. indirect estimates of $U(Y : X_2 \backslash X_1)$, illustrating the consistency of the method.

# G  More on the experiments

In this section we provide more details on the last two experiments presented in Section 4.

**Computational aspects of connectivity in recurrent neural circuits.**
We start by deriving the relation $TE = S + U_1$ verified in this experiment (Fig. 3d). Transfer entropy [52] $TE(X \rightarrow Y)$ is defined as $I(Y^+ : X^- \mid Y^-)$ where $Y^+$ is the future of state of $Y$, $X^-$ and $Y^-$ are the past states of $X$ and $Y$, respectively. Consider the chain rule for mutual information,

$$I(Y^+ : (X^-, Y^-)) = I(Y^+ : Y^-) + I(Y^+ : X^- | Y^-). \tag{G.1}$$

Replacing $I(Y^+ : (X^-, Y^-))$ and $I(Y^+ : Y^-)$ by the r.h.s. of (1.1) and (1.3), we get

$$I(Y^+ : X^- | Y^-) = U(Y^+ : X^- \backslash Y^-) + S(Y^+ : (X^-, Y^-)), \tag{G.2}$$

| N | K=4 | | | | K=6 | | | | K=8 | | | | K=10 | | | |
|---|---|---|---|---|---|---|---|---|---|---|---|---|---|---|---|---|
| | UI | SI | MI | W | UI | SI | MI | W | UI | SI | MI | W | UI | SI | MI | W |
| 1 | 0.28 | 0.27 | 1.22 | -0.05 | 0.72 | 0.38 | 1.1 | -0.11 | 0 | 0.22 | 1.59 | 0.09 | 0.05 | 0.19 | 2.7 | 0.27 |
| 2 | 0.39 | 0.59 | 0.9 | -0.57 | 0.01 | 0.39 | 1.24 | -0.15 | 1.58 | 0.42 | 2.93 | -0.57 | 1.74 | 0.13 | 3.47 | -0.55 |
| 3 | 0.04 | 0.39 | 0.84 | 0.18 | 0.01 | 0.39 | 0.93 | -0.14 | 0.04 | 0.05 | 1.18 | 0.07 | 0.38 | 0.46 | 2.21 | 0.04 |
| 4 | 0.01 | 0.22 | 1.2 | -0.38 | 0.02 | 0.32 | 0.93 | 0.02 | 0.34 | 0.33 | 2.16 | 0.03 | 0 | 0.37 | 1.24 | 0.02 |
| 5 | 0.4 | 0.29 | 1.38 | -0.34 | 0.56 | 0.69 | 1.8 | -0.43 | 0.01 | 0.41 | 3.39 | -0.62 | 0.02 | 0.2 | 2.32 | -0.21 |

Table 1: **Node-specific details for generalized XOR task:** Average node-specific unique, synergistic, and mutual information (UI, SI, MI) and the decoding weight for different nodes in the hidden layer ($N \in \{1, \ldots, 5\}$) and for different task difficulty levels ($K \in \{4, 6, 8, 10\}$).

which is the equation we verified by estimating separately the left and right sides. The two terms in the r.h.s. are called *state-independent transfer entropy* and *state-dependent transfer entropy* respectively in [55], reflecting their intuitive meaning.

In Fig. 7, we analyze the state space of the network in Fig. 3 of the main text. The activities of the upstream sub-network X and downstream sub-network Y are shown, projected onto their first two principal components (PCs). The causal structure and algorithmic details of the effective connectivity between the two sub-networks cannot be identified solely by the observation of their geometrical properties.

**Uncovering a plurality of computational strategies in RNNs trained to solve complex tasks.**
Each RNN has fully connected architecture with `tanh` non-linearity. Data was generated by sampling from the GMM with K components ($K \in \{4, 6, 8, 10\}$) in batches of 128 data points with the total number of 3000 batches. The RNNs were trained using standard backprop in time using `Adam` optimizer in `Pytorch` package with a learning rate of 0.01. For each trained RNN we considered all triplets $(Y, X_i, X_j)$ where $i, j \in \{1, \ldots, 5\}$, i.e. the target variable is the output of the network $Y$ and the source variables iterate over all pairs of the hidden nodes in the RNN. Once the RNN is trained we collect a test sample of 1000 data points from the same GMM used for training, and evaluate the nodes when inputting the RNN using test data and running it forward for $t = 10$ time steps. This gives us 1000 samples from each variable $X_{1:5}, Y$ which we then use for PID analysis on the triplets mentioned above. For each level of task difficulty $K \in \{4, 6, 8, 10\}$ we trained 5 RNNs and performed PID ($A = 100$) on the resulting trained networks.

In Fig. 8 more details on the trained RNN's in Fig. 4 of the main text are illustrated, providing more insight into the computational strategies employed by each trained instance as the task complexity grows. The first row shows the time evolution of the recurrent layer of hidden units projected onto their first 3 PC's. For these RNN instances, the ones with $K = 6, 10$ have grand mother-like cells (as confirmed by the receptive field plots in Fig. 8c), with large unique information compared to the other cells. These grand mother-like cells cannot be inferred by just inspecting the geometry of the hidden units in the state space, but can be identified with the PID. PID reveals more details about the computation and the differences between strategies for different instances of trained RNN's. Details of the PID for individual hidden nodes including average unique and synergistic information for each node, its mutual information with the output node, and the decoding weight connecting the hidden node to the output unit is included in Table 1.

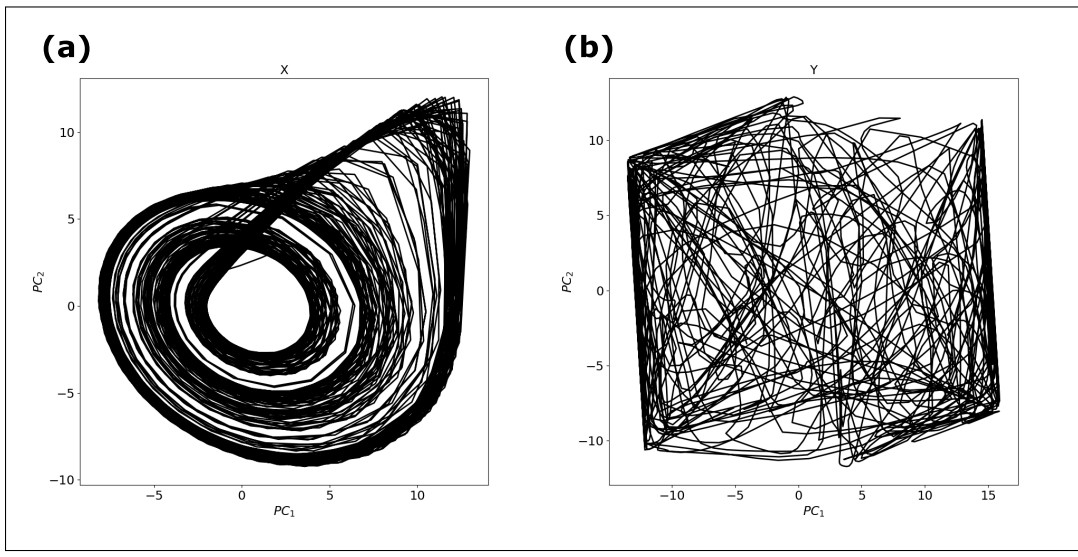

Figure 7: **State space of the chaotic network of rate neurons:** Projection of the state space of the recurrent units for upstream network $X$ (a) and downstream network $Y$ (b) onto their respective first two principal components.

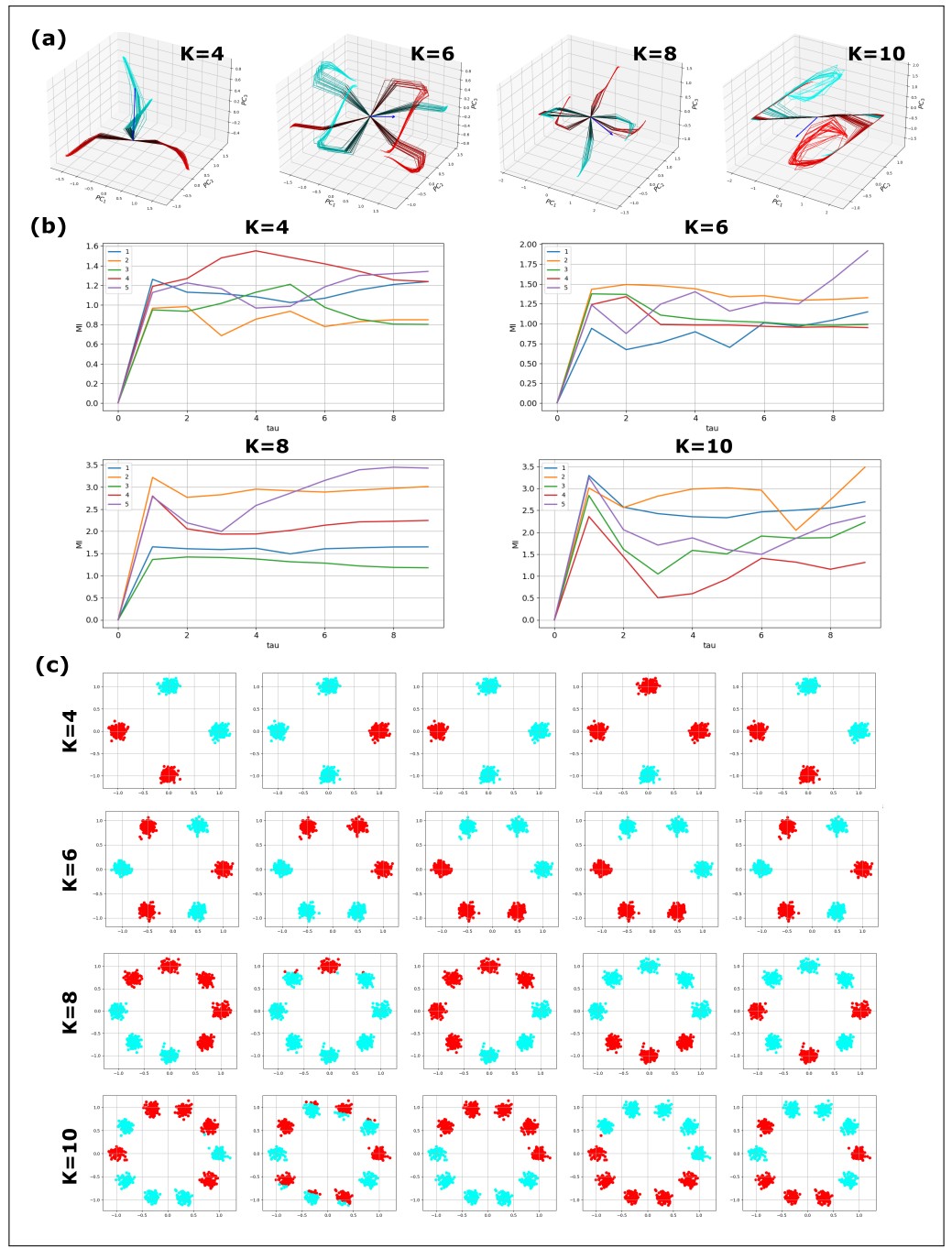

Figure 8: **Algorithmic investigation of trained RNN's on generalized XOR task:** (a) Evolution of the hidden unit activations in time (recurrent time steps). Darker colors correspond to earlier time points; red and cyan correspond to even and odd trials. Blue arrow corresponds to decoding direction, i.e. the predicted label is given by the sign of the projection of the last time point of each trajectory onto this direction. (b) Mutual information between individual hidden units and the output of the network as a function of recurrent time steps for the different tasks. (c) Receptive fields of individual neurons, in certain cases (K=6, unit 1 and K=10, unit 2) grand mother-like cells can be observed, yielding greater unique information than synergistic information hinting at the algorithmic strategy employed by that instance of the trained RNN.