# OpenReview forum: "Estimating the Unique Information of Continuous Variables"
_NeurIPS.cc/2021/Conference — NeurIPS 2021 Poster_

### Official Review · Reviewer_7Ede · 2021-07-13

**Rating:** 7
**Confidence:** 2

**Summary:**

(I am not an expert in information theory)
The authors extend partial information decomposition to the case of continuous variables with arbitrary distributions.
My review will focus on the last section, where a recurrent neural network was trained on a generalized XOR task. By decomposing the unique, redundant and synergistic components of each neuron’s contribution to the output, the authors identify different strategies used by these networks.


**Limitations And Societal Impact:**

not relevant

**Main Review:**

I am not an expert in information theory, and will therefore focus on the last section – studying a recurrent neural network trained on a generalized XOR task.

The question of algorithm identification in trained networks is both important and challenging. One aspect of this challenge is the difficulty to define what it means to identify an algorithm. Is it some connectivity motif? An arrangement of fixed points? The geometry of phase space trajectories? The response to novel inputs?

Given this state of affairs, novel suggestions on how to characterize algorithms can be useful. In the specific example given here, however, the authors did not clearly define what they mean by algorithm. The analysis shows that the properties of individual neurons differ both within networks and across networks. Specifically, some single neurons in the recurrent layer seem to already encode most of the desired solution to the XOR problem.

In terms of an algorithm, I find this description lacking. How do these neurons obtain such a representation? By definition, the output of the network will have this property. Finding such internal neurons merely moves the question, but does not answer it.

Furthermore, this notion is very similar to that of mixed selectivity (Fusi et al. Curr opinion neuro, 2016). This might imply that linear dimensionality, which is much simpler to compute, could provide similar information on the networks.

I also found figure 7C puzzling. For K=6, both neurons 1 and 5 seem to have very similar receptive fields and can be described as grandmother cells. Neuron 5 has a higher mutual information, and neuron 1 has a higher unique information. It could be useful to use this case to explain the differences and how they originate.

Furthermore, the networks analyzed here are very small (5 neurons) compared to those cited in line 227 (100 neurons). How does this method scale?

Overall, this specific example does not demonstrate the utility of this method in inferring algorithms from trained networks.

Once again, I would like to stress that this review only relates to one small aspect of the paper.


**EDIT: Score raised from 4 to 7 after reading all reviews and author responses.**



**Time Spent Reviewing:**

1

---

> ### Author Response · Authors · 2021-08-09
> **semantics of 'algorithm identification' and relationship with mixed selectivity**
>
>
> We thank the reviewer for the time devoted and the interesting comments on the generalized XOR task.
>
> The reviewer recognizes the important challenge of algorithm identification in trained networks, expresses his concern regarding our lack of a technical definition for the word ‘algorithm’ and suggests that the PID results presented fall short of the expression "algorithm identification".
>
> Unlike the possibilities the reviewer mentions, stemming from the fields of network science or dynamical systems, the PID analysis makes no claims on the underlying computational mechanisms. On the contrary, a level of abstraction from such mechanisms is a hallmark of information-theoretic characterizations of complex processes. In this particular case, the PID analysis provides a valuable lens to neural activity, by showing that the presence of 'grandmother' neurons (which cannot be inferred by inspecting the geometry of the hidden units in the state space) is consistently captured by high unique information. It is in this abstract information-theoretic sense that we used the expression "algorithm identification", but we are happy to adopt an alternative description, perhaps better aligned with the reach of the PID analysis.
>
> We thank the reviewer for suggesting a comparison between our PID-based approach and the concept of neuronal selectivity used in neuroscience. We first summarize the concept of mixed selectivity and then clarify the differences between the approaches.
>
> Neuronal selectivity is a property of single neuron responses to multiple task variables. E.g., in a decision-making task where a monkey saccades to the left (right) when presented with a red circle or a blue square to obtain a reward, then the two task variables could be the color (red vs blue) or the shape (circle vs square). Several kinds of selectivity can be identified by performing regression analysis of neural responses vs task variables. A neuron has pure selectivity if it is selective to a single task variable (e.g., colors, with large responses to red and no responses to blue), but no other variables (e.g., no responses to shape) - like a grandmother cell. It has linear mixed selectivity if it is selective to both colors and shapes in a linear combination. Both pure and linear mixed selectivity properties can be detected with a linear regression of the neuron’s activity vs. the task variables. On the other hand, the neuron has nonlinear mixed selectivity if its responses to colors and shapes yield an interaction term in the regression analysis. The approach of Fusi et al. (and of Rigotti et al., Nature, 2013) applied to the XOR task shows that non-linear mixed selectivity of neurons to the input class labels is beneficial when solving the XOR task, by leading to a high-dimensional representation of the task variables.
>
> Now, in our experimental results we estimated the PID between the activity of pairs of hidden layer neurons and the output neuron, as opposed to the selectivity of single neurons to input stimuli as in Fusi et al. Crucially, while selectivity profiles are a property of *single neuron* responses to task variables, our PID measures are a property of the combined activity of *triplets of neurons* and thus reveal emerging functional interactions between units and their computational algorithms. This allowed us to characterize a functional property of neural systems less studied than task variable selectivity: the computations that require functional mixing of the information from multiple units (measured by the average synergistic information) vs. the computations that rely on the output of individual neurons (measured by the unique information and described as grandmother cells). A similar approach was previously used by Timme et al (PLOS 2015) based on discrete PID showing that neurons that compute large amounts of information tend to receive connections from high out-degree neurons. In other work by Wibral et al (Entropy 2017) redundant information was linked to robust coding in neural systems.
>
> Concretely, in Figs. 4 and 7 we first computed the PID for neuronal triplets, and then investigated the selectivity properties of single neurons to task variables as encoded in their receptive fields. Our PID approach revealed that recurrent networks can solve the generalized XOR task by employing a variety of alternative strategies. In networks with high unique information, neurons typically have receptive fields with pure selectivity (grandmother cells, with large unique information to the task variables). In networks with high synergy, neurons do not show pure selective but rather complex mixed selectivity profiles with respect to task variables. Although a full characterization of the relation between PID and mixed selectivity is beyond the scope of the current work, we thank the reviewer for highlighting this tantalizing direction for future investigation and we plan to elucidate this issue in future work.
>
> The reviewer asks about the difference between neurons 1 and 5 for the case K=6 in Figure 7c. While both neurons are grandmother cells, the reviewer wonders why neuron 5 has a higher mutual information MI, and neuron 1 has a higher unique information U. The reason for this is very simple: the values of U and MI assigned to neuron i are the mean of all the values of U and MI estimated over all the possible neurons j in the triplet (neuron i, neuron j, output). Therefore, we expect to see variability in the estimates of MI and U across different neurons, even among sets of grandmother neurons. The important message here is that the unique information of non-grandmother units (neurons 2,3,4) is close to zero.
>
> Regarding scaling to higher numbers of neurons, the computational cost of this PID analysis scales quadratically with the number of hidden units, since the estimates presented require averaging over PID values of all pairs of hidden units.
>
> We hope the above comments encourage the reviewer to take a fresh look at the paper and revisit the negative rating.

---

> > ### Comment · Reviewer_7Ede · 2021-08-22
> > **Changed rating**
> >
> > I thank the authors for their response, and for clarifying the analysis done on the RNN example.
> >
> > If I understand correctly, the apparent discrepancy between tuning and S/U/R values in figure 7 stems from the averaging procedure: The analysis is on how pairs of hidden units combine to affect the output unit. But the results presented average over this and report the average S/U/R values for each single neuron. Is it possible to utilize this pairwise (I know it's triplets, but the third one is always the output) information better? For instance: eigenvectors and eigenvalues of the S/U/R matrices. Perhaps more.
> >
> > In any case, given the other reviews and author responses, and the fact that this RNN example is not the center of the submission, I am raising my score to a 7.

---

### Official Review · Reviewer_BEa7 · 2021-07-16

**Rating:** 8
**Confidence:** 4

**Summary:**

   This paper presents a novel method for computing the Partial
   Information Decomposition (BROJA flavour) in the case of three
   continuous random variables. The method uses a variational approach
   on a lower bound derived from a copula representation of the joint
   PDF of the three variables. The paper presents some application
   examples, including two where the proposed method is used to
   elucidate the properties of recurrent neural networks.

**Limitations And Societal Impact:**

The authors have adequately addressed the limitations and potential negative impact of their work.

**Main Review:**

This paper addresses an important issue standing in the way of
practical application of PID to the analysis of real data. To the best
of my knowledge, the approach is novel, and I could not identify any
formal or technical problem. The paper is well written and for the
most part does a good job of describing the relationship of this work
to existing literature. I particularly appreciated the very accessible
summary of the decision-theoretic foundation of the BROJA PID. Overall
this is in my opinion a good and useful paper that would benefit the
NeurIPS community.

Having said the above, I would like to bring two points to the
attention of the authors.

1. While reading the paper, I got the distinct impression that the
   title of the paper is somewhat misleading, in the sense that to me
   the "meat" of the paper is clearly the novel PID estimation method,
   while the recurrent network side is more of a demonstration of
   potential usefulness. I was confused by this until I happened to
   see the submission history of the paper (that is, that the RNN side
   was added partly to address previous criticisms). While I think
   that the RNN results are interesting, I am of the opinion that the
   paper would be more likely to reach an interested audience / more
   likely to have a bigger impact if the title was changed to
   something more related to the PID itself (as I suspect that the
   original title must have been in the previous submission). In the
   same vein, the neural-networks-first presentation in the
   introduction may be softened.
2. I couldn't find a link to a software implementation of the proposed
   technique. I don't think this is a hard requirement, but it would
   certainly be of great help for the interested reader, even if it is
   just unpolished set of functions/scripts that allow to at least to
   reproduce some of the results in the paper.

Minor points:
- Lines 86--87: "PID has been previously applied to in vitro neural
  data to elucidate how single neuron computational properties depend
  on network topology [ref to Timme et al, PLoS comp bio 2016]". If I
  am not mistaken, the PID in Timme et al is based on the old Williams
  and Beer approach. By contrast, the method proposed by the authors
  in the current paper is based on the three-variable BROJA PID. An
  early application of the BROJA PID in neuroscience is Pica et al,
  NeurIPS 2017, "Quantifying how much sensory information in a neural
  code is relevant for behavior.", so the authors may consider adding
  this reference too.
- Line 187: missing period at the end of the paragraph.


**Time Spent Reviewing:**

3

---

> ### Author Response · Authors · 2021-08-10
> **Title and link to code**
>
> We thank the reviewer for the positive feedback and the comments.
>
> -**Title**: We agree that including 'recurrent networks' in the title might give the wrong impression that the method is limited to RNNs. We will revise the title accordingly, omitting any reference to recurrent networks.
>
> -**Code**: python/pytorch code to reproduce the results of Figure 2 is now available in the link indicated in Appendix G (https://tinyurl.com/pidneurips)
>
> We will also add the suggested reference and thanks for catching the typo.

---

> > ### Comment · Reviewer_BEa7 · 2021-08-24
> > **Response to authors**
> >
> > Thank you for addressing the points I raised. I hope that these changes will help the paper reach a broad audience and achieve the impact it deserves.

---

### Official Review · Reviewer_ToUB · 2021-07-22

**Rating:** 7
**Confidence:** 4

**Summary:**

This paper extends the popular PID measure of Bertschinger et al. (2014) to continuous random variables, and provides a method to compute and estimate the PID in this case. The authors use a novel method based on copulas to reparameterize the optimization problem used to compute the PID, and estimate it by using an approximation based on a variational upper bound. They also provide several examples to show that their estimates match known PID quantities and that the PID can be useful in various neuroscientifically relevant settings.

**Limitations And Societal Impact:**

Limitations have been adequately discussed.

**Main Review:**

Overall, the technical section of the paper remains hard to read for an audience unfamiliar with copulas, despite the efforts of the authors to introduce these concepts. The notation is often times confusing and challenging to parse. However, I believe the paper is of substantial significance, since no methods to compute or estimate PIDs on general continuous distributions currently exist. As the experimental results outlined in the paper illustrate, such a method could be of great interest in several application domains, particularly in neuroscience.

Originality:
The paper is very original, since there are no methods to estimate PIDs for general continuous distributions. Most works discuss computation of the PID, assuming full knowledge of the distributions, rather than estimating the PID, which requires the additional step of inferring distributions from data.

(Technical) Quality:
Barring the issues with clarity of presentation and notation (which are discussed in detail below), I believe the paper is of high technical quality: the authors have used a number of concepts and have been thorough in their references: they provide a good sampling of the relevant background and literature for each concept. The idea of using copulas for the Bertschinger et al. PID is intuitively appealing.
I have one technical question: in Figure 1, why is there a large difference in the left bottom corner? The left-bottom should be where the question is easiest: $\rho_{Y, X_2}$ is close to 1 and $\rho_{Y, X_1}$ is close to zero, so it should be easy to determine that the unique information (in $X_1$) is zero.

Clarity:
I believe the technical section can be significantly improved in its presentation:
1. It would be useful to have an short outline at the start of Section 3 briefly describing the steps involved in estimating the unique information. Otherwise, it is hard to see what the various sub-parts are leading towards.
2. In equation 3.3, is $c(u_y, u_2)$ the distribution that results from summing $c(u_y, u_1, u_2)$ over $u_1$? Please clarify.
3. Line 107: What does the term "copula degrees of freedom" mean? This has not been introduced yet.
4. Equations 3.5 and 3.8: it is not immediately obvious how the conditional distribution can be written in terms of the copula. An extra line explaining this near Equation 3.1 may be useful.
5. Line 111: "Note that the function ..." - it might be useful to have this sentence appear a little earlier, just after Equation 3.8.
6. Line 119: "parameterizing by $\theta$"  - it remains unclear after this sentence exactly what $\theta$ parameterizes. If $\theta$ is the same as $Q$, why not use that instead?
7. Line 121: $\theta(y)$ is now being used as a function. Please clearly explain what $\theta$ is.
8. Equation 3.14: $I_q[\theta]$ is confusing notation, since $q$ does not appear in the RHS. Also, the second term in the RHS implicitly converts an expectation over three variables to one over two. This is only valid if summing $c_{\theta}(u_y, u_1, u_2)$ over $u_y$ is equal to $c_{\theta}(u_1, u_2)$.
9. Line 128: Does the $D \log D$ cost come from sorting the variables for computing the CDF? Please clarify.
10. Please mention how long the estimation algorithm took to run, and on what hardware. Running time is an important aspect of the reusability of this work.
11. Line 214: "We treated ... as iid samples and ran PID on these triplets" - why is this a sensible thing to do for the question being explored? Please explain briefly.

Minor comments:
1. The negative vspace after title should be removed.
2. Line 81: "exits" should be "exist"
3. Figure 3b: Bottom-right quadrant has $X_t$ in place of $Y_t$
4. Line 251: "... PID framework abstracts _and_ generalizes ..."

Significance:
I believe this paper is very significant: it provides the first usable PID estimator for general continuous random variables, and is quite likely to be used by a number of works seeking to apply PIDs in different contexts.

**Update after author responses**

Based on the authors' responses, I am happy to fully recommend accepting the paper, and I have raised my score accordingly.

**Time Spent Reviewing:**

6

---

> ### Author Response · Authors · 2021-08-06
> **a technical question and comments on clarity**
>
> We appreciate the significant time devoted by the reviewer to a thorough reading of the paper and the detailed technical comments. We will certainly include this feedback in the final version.
>
> Regarding the question on Figure 1, our present hypothesis is that the behavior of the error in the left bottom corner might be due to the particular parametrization we used for the copulas. This is the type of bias that we mentioned in Lines 166-167. We plan to explore this point more thoroughly and report any conclusive finding in the final version.
>
> We respond point by point to the comments on clarity:
>
> 1. Thanks for the suggestion. We agree that a short outline at the start of Section 3 would increase the overall clarity and we will incorporate it.
>
> 2. Indeed. After defining the joint copula density as $c(u_y,u_1,u_2)$, we indicate marginal densities with the same symbol $c(\cdot)$. Thus, we have $c(u_y,u_2) = \int du_1  c(u_y,u_1,u_2)$ and $c(u_1,u_2) = \int du_y  c(u_y,u_1,u_2)$. For the latter case, this is mentioned in Lines 133-134, but we will make these identities explicit for clarity.
>
> 3. We used the expression "degrees of freedom" in line 107 in an informal sense, in order to convey the idea that even after Proposition 1 shows that the unique information only depends on the copula, the latter can be further decomposed into (i) factors that stay constant and (ii) factors that change as one moves in the space of probabilities $Q$ defined in (1.7). "Degrees of freedom" was meant to refer to case (ii), as those factors are 'free' to change inside $Q$. But since the expression "degrees of freedom" is commonly used in a more quantitative sense in other settings, we will replace it in order to avoid confusions.
>
> 4. We will add intermediate steps to make the equations obvious. So equation (3.5) will become
> \begin{align}
> p(y|x_1) = \frac{p(y,x_1)}{p(x_1)} =  \frac{ p(y) p(x_1) c(u_y, u_1)   }{p(x_1)} = p(y) c(u_y, u_1)
> \end{align}
> and equation (3.8) will change similarly.
>
> 5. Agreed, we will fix that.
>
> 6.  To clarify the notation of $\theta$ and $\theta(y)$, let us denote by $\theta$ the parameters of the bivariate copula associated with $(x_1,x_2)$. We can indicate this as
> $$p(x_1,x_2) = p(x_1)p(x_2)c_{12|\theta}(u_1,u_2).$$
> One way consider the conditional distribution $p(x_1,x_2|y)$ is to incorporate $y$ as a "parameter" in each factor of the above decomposition, leading to
> $$p(x_1,x_2|y) = p(x_1|y)p(x_2|y)c_{12|\theta(y)}(u_{1|y},u_{2|y}).$$
> In particular, the parameters $\theta$ become a function of $y$. Since we show in Proposition 2 that the space Q is parametrized by the copula of the conditional, it follows that the space Q is parametrized by the the function $\theta(y)$, whose parameters are also denoted by $\theta$ in a slight abuse of notation. We hope this clarifies the notation, and we will clarify this point similarly in the final version.
>
> 7. See point 6 above.
>
> 8. The subscript $q$ in $I_q[\theta]$ was meant to indicate that the objective function in (3.14) is the same as those in (1.5) and (3.3), which depend explicitly on the distribution $q \in Q$. But we agree that this leads to a confusing notation, so we will drop $q$ from (3.14). The second point (on marginalization) was answered above in point 2.
>
> 9. Indeed, in order to convert D points in the $(y,x_1,x_2)$ space to D points in the CDF space $(u_y,u_1,u_2)$, the original points must be sorted, and thus the O(D log D) cost. This will be made clear.
>
> 10. For a dataset of 3000 points $(y,x_1,x_2)$, estimation of the four terms (U1, U2, S, R) of the PID takes about 11.4 seconds on a Linux PC Intel i5 @2.90GHz with GeForce RTX 2070 SUPER.
>
> 11. The mention of 'iid samples" in Line 214 points out that the estimation of the PID from many samples of the triplets $(Z_i^t, Z_j^t, Z_j^{t+1})$ is operationally the same whether such samples are iid or, as in our case, temporally correlated. This is similar to estimating expectations w.r.t. the equilibrium distribution $p(x)$ of a Markov chain $x_{t-1} \rightarrow x_{t}$ by using temporally correlated successive values of $x_t$. In both cases, the temporal correlations do not introduce bias in the estimator but can increase the variance.  We will make this point clear in the text.
>
>
>
> Thanks also for the 'Minor comments', we will fix those typos (just note that we did not add any negative vspace after the title).

---

> > ### Comment · Reviewer_ToUB · 2021-08-19
> > **Response to authors**
> >
> > I thank the authors for their detailed response to my questions. I feel all of my concerns have been adequately addressed.
> >
> > I encourage the authors to look into the issue of bias in Figure 1 more closely, and acknowledge this point in the limitations of their work. As I had mentioned in my review, I feel this should be a situation where it should be easy to tell that the unique information is zero (based on the result of [30] for 1D gaussian variables).

---

### Official Review · Reviewer_VXUZ · 2021-07-23

**Rating:** 7
**Confidence:** 2

**Summary:**

Authors present a partial information decomposition (PID) method that can be applied to continuous variables thereby extending a popular approach for analysis of discrete variables. Authors validate their method  by leveraging known analytic results for Gaussian random variables. Additionally, the authors demonstrate that their approach is able to recover the effective connectivity of a neuronal network with chaotic dynamics. Finally, authors use their method to investigate a complex trade-off between redundant, synergetic, and unique information components in recurrent networks that have been trained to solve the XOR problem.

**Limitations And Societal Impact:**

Yes

**Main Review:**

The algorithm presented in this manuscript is derived from an existing one for discrete variables, but the authors manage to extend the method to include continuous variables, thereby producing a novel and widely applicable technique. There are no obvious flaws that I can see in the theory and experiments presented in this paper. However, I did not review the theory in detail and it is not impossible that I have missed many important details. The paper is written clearly enough and provides enough experimental evidence to back the main claims introduced by the authors. PID for continuous variables can have a number of applications (specially in the field of neuroscience) as indicated by the authors and therefore the contributions presented in this manuscript are quite significant. An interesting addition to this paper would be to add an example application of the method using real data (from the brain), but I do not see this as requirement for this manuscript to be accepted in the conference.

**Time Spent Reviewing:**

2 hours

---

> ### Author Response · Authors · 2021-08-06
> **on real neural data**
>
> We thank the reviewer for the time devoted and the comments.  Indeed, a major application area of our method is real neural data. While we might add some preliminary results to the appendix, we feel that a proper discussion would exceed the scope of the present paper.

---

### Decision · Program_Chairs · 2021-09-27

**Decision:**

Accept (Poster)

**Comment:**

This work presents a new method for partial information decomposition that was found to be novel and interesting. Given the narrowing of the scope (away from the RNN application and more focused on the PID estimation) proposed by the authors in response to the reviewers initial comments, the reviewers agreed that the work was relevant to the NeurIPS community and that the method and applications would be greatly appreciated. I thus am happy to recommend this paper be accepted.